# Social inequality in infant mortality in Angola: Evidence from a population based study

**Gebretsadik Shibre**📵*

Department of Reproductive, Family and Population Health, School of Public Health, Addis Ababa University, Addis Ababa, Ethiopia

* gebretsh@gmail.com

## Abstract

### Introduction

Within country inequality in infant mortality poses a big challenge for countries moving towards the internationally agreed upon targets on child mortality by 2030. There is a lack of high-quality evidence on infant mortality measured through different dimensions of social inequality in Angola. Thus, this paper was carried out to address the knowledge gap by conducting in-depth examination of infant mortality rate (IMR) inequality among population subgroups to provide more nuanced evidence to help end IMR disparity in the country.

### Methods

The World Health Organization's (WHO) Health Equity Assessment Toolkit (HEAT) was used to analyze IMR inequality. HEAT is a software application that facilitates examination of disparities in reproductive, maternal, neonatal and child health indicators using the WHO Health Equity Monitor (HEM) database. Inequality of IMR was analyzed through disaggregation by five equity stratifiers: education, wealth, gender, subnational region and residence. These were analyzed through three inequality measures: Population Attributable Risk, Ratio and Slope Index of Inequality. A 95% confidence Interval (CI) was built around point estimates to determine statistical significance.

### Results

A notable disadvantage was found for children born to poor (Population Attributable Risk (PAR): -27.0; -28.4, -26.0) and uneducated (PAR: -17.0; -17.9, -16.0), women who live in rural areas (PAR: -7.3;-7.8, -6.7) and those residing in certain regions of the country (PAR: -43.0; 45.3, -4). Male infants had a higher risk of death than female infants (PAR: -6.8;-7.5, -6.2). The subnational regional variation of IMR had been the most evident when compared with the disparities in the other equity stratifers.

### Conclusions

Policymakers and planners need to address the disproportionately higher clustering of IMR among infants born to disadvantaged subpopulations through interventions that benefit such subgroups.

**Data Availability Statement:** The datasets generated and/or analyzed during the current study are available in the WHO's HEAT version 3.1 [https://www.who.int/gho/health_equity/assessment_toolkit/en/]. HEAT, Built-in Database Edition.

**Funding:** The author(s) received no specific funding for this work.

**Competing interests:** The authors have declared that no competing interests exist.

## Introduction

Infant mortality rate (IMR) is a measure of the number of infant deaths per 1000 live births born to a group of women in a specified time period [1]. Improving lives of new born babies and increasing their survival advantage has been one of the main responsibilities of social policies [2]. Intervention that aim to improve survival of infants target problems such as birth defects, preterm birth, low birth weight, pregnancy related complications, sudden infant death syndrome, and injuries [1]. Not only does IMR offer useful information about infant health [1], it remains an imperative marker of a community's health, supporting the assertion that problems impacting health of an entire population cause measurable influences on the mortality of infants [1, 3]. For instance, IMR was one of the indicators used to measure progress towards the "health for all by 2000" in 1981 [4].

Despite remarkable progress over the last several years globally, infant mortality still remains a major public health problem. In 2018, four million infants died, accounting for more than three forth of the global under five mortality burden [5]. However, glaring variations in IMR remain between countries and regions worldwide, with the highest burden concentrated in Sub-Saharan Africa (SSA) [6]. Research on the country level IMR demonstrated that the highest IMR was reported in Afghanistan with 110.6 deaths per 1000 live births, followed by Somalia with 94.8 deaths per 1000 live births and the lowest was in Monaco with just 1.8 deaths per 1000 live births [7] Angola ranked 12[th] with 67.6 deaths per 1000 live births, and has the highest IMR burden globally.

Evidence has shown between country variations of IMR as well as perceptible within country disparities across different dimensions of inequality. Infant mortality varied significantly based on where infants were born and where they live [8–10]. Systematic differences in the clustering of problems that cause infant mortality and in the health care services for infant population between geographical locations were mentioned as drivers of geographical variation of IMR [9]. Women's economic status appeared to influence deaths during the infantile period, and it has been shown that infant moralities are more concentrated among poor families than among the richest families [11, 12]. Though in some countries income gradient in infant mortality had not been observed, some literature found that infant mortality does not affect the poor and the rich equally [12]. Moreover, evidence that disparity in socioeconomic status between population groups could lead to inequality in distribution of infant mortality according to the different social and economic classes [13] signals the importance of implementing pro-poor policies to eliminate income and education driven disparities in infant mortality. Socioeconomically disadvantaged subgroups in low-and-middle income countries contribute a disproportionately higher concentration of infant deaths compared to higher income subgroups [14]. Interestingly, the unequal distribution of infant mortality in a country has not been confined to just geographical location and socioeconomic status. Prior work demonstrated a large gender differential of IMR [15–17] with male infants enduring a higher burden of death than female infants.

Located in the west coast of Southern Africa, Angola has one of the poorest health-care systems. The infant and child mortality rates in the country are unacceptably high compared with that of other countries in the SSA. A large information gap on health care indicators makes it difficult to inform the decision making process in the country [18]. Furthermore, Angola is lagging behind on health equity and financial protection indicators as demonstrated by the large inequality in health outcomes and health care utilization between different population groups [19]. Identifying areas of inequalities has been among the major priorities of Angola's health sector [20] in order to redress the currently prevailing health disparity in the country. The Millennium Development Goals (MDG) have been criticized for relying on aggregate

indicators and for inattention to internal inequalities within countries. On the other hand, equity has been one of the hallmarks of the Sustainable Development Goals (SDG) [21] and evidence on state of inequality of health care indicators is important to track improvements of health disparities between now and 2030. Evidence suggests that substantial within country disparity in IMR signals widespread problems in basic sanitation, health care services, nutrition, and education [1], supporting the assertion that combating within country inequality in IMR could result in huge health gains and returns.

Drawing on the available literature on IMR disparity, the present study has extended the evidence in many ways. First, there are no studies in Angola that provided in-depth interpretation of the IMR disparities by the five equity stratifers (wealth, education, residence, gender and regions). Second, the present study follows the WHO recommendation for inequality analysis [22] so that findings can inform equity interventions to end health inequalities. This paper intended to examine the extent of IMR variations between different population subgroups in Angola and to assess the likely impact of the variation on the level of IMR nationally.

## Methods

### Data source

The offline version of the WHO HEAT software was used to analyze the data; the software has been described in detail elsewhere [23, 24]. Briefly, the HEAT software allows the examination and in-depth descriptive analysis of health care indicator inequalities within and between countries of more than 30 reproductive and maternal health care indicators. The WHO HEAT contains the WHO Health Equity Monitor (HEM) database [25] that stores data derived from Demographic and Health Surveys (DHS) and Multiple Indicator Cluster Surveys (MICS). The database comprises of data collected through household surveys that have been carried out in many low-and-middle income countries including Angola.

The 2015 Angola Demographic and Health Survey (ADHS) available in the HEAT software were used for this study. The 2015 ADHS, a household survey, collected data on various health and demographic indicators such as maternal health services, child health, maternal and childhood mortality, family planning and domestic violence. Information obtained through the survey is used to monitor the health status of the population in Angola. Samples were representative at the national, provincial, urban, and rural levels as well as for sociodemographic characteristics such as sex, age, education and household wealth. A total of 16,244 households were selected, of which 16,109 were interviewed, yielding a response rate of 99%. Among the interviewed households, 14,975 women aged 15–49 years were identified and 14,379 were interviewed, resulting in a response rate of 96%. The survey focused on women age 15–49, but data were also gathered from men age 15 to 49 and children under five. The National Statistics Institute cooperated with the Ministry of Health (MINSA) to implement the survey, with technical assistance from United Nations Fund for Children (UNICEF) and Inner City Fund (ICF) International through the DHS Program and the WHO. United States Agency for International Development (USAID), through the President of the United States Initiative for Malaria Control (PMI) and the United States President's Emergency Plan for AIDS Relief (PEPFAR), the World Bank through the Health Municipalization Program, UNICEF and the Government of Angola provided financial support. The survey was carried out from October 2015 to March 2016.

Methodology of the 2015 ADHS has been discussed in detail in the survey's final report [26]. For the purpose of sampling, the land of Angola was divided into Census Sections (SC). Sampling frame of the Primary Sampling Units (PSU) was prepared from the 2014 Angola Population and Housing Census (PHC) and stratified by province as well as urban and rural

areas from which the mother sample was selected. The ADHS used a stratified three-stage cluster sampling technique to draw samples. In the first stage, PSUs were selected systematically with Proportional Probability to Size (PPS) within each stratum. In the second stage, within each stratum, Secondary Sampling Unit (SSU) was selected through PPS sampling technique. The final sampling stage was the random selection of 26 households listed in each SSU. Households eligible for the interviews were selected with equal probabilities within the SSU.

## Variables of the study

IMR is the primary variable of the study and is defined as the number of deaths before celebrating first birth day per 1000 live births. The survey collected full birth histories for women aged 15 to 49 years and children (including date of birth and age of death). Children born 5 years preceding the survey were included in the analysis. The inequality in IMR was examined using five common dimensions of inequality: economic status, educational status, sex, subnational regions and the place of residence.

## Dimensions of inequality

Unlike the case for health monitoring, where measuring health indicator variables is enough, studying social inequality in health requires variables related to the health indicators of interest across equity stratifiers. The WHO defines equity stratifiers as dimensions of inequality by which a health indicator is to be disaggregated [22]. Social inequalities point to health inequities and unjust disparity in health between social groups if the cause of the social inequality is avoidable. Preferably, health inequality should be analyzed and presented using all dimensions relevant for the health indicator in question. For many years, attention has focused health inequality in terms of economic status. However, the WHO has developed other policy-relevant dimensions of inequality using equity stratifers that include: place of residence (rural, urban, etc.), race or ethnicity, occupation, gender, religion, education, socioeconomic status and social capital or resources.

In this paper, IMR inequality was measured by the five equity stratifiers of economic status, education, residence, subnational region and gender. The choice of dimensions of inequality was based on their relevance to IMR as well as availability of IMR data for each of the stratifiers. Economic well-being of households was measured through a wealth index which was computed based on different physical durable assets owned by households and on features of the household dwelling. While variables used for creating wealth index differ between surveys [27], such features and assets as water and sanitation facilities (WASH), radio, television, types of materials used to make floor, roof and wall of a household, car, bicycle, motorcycle, and electricity have been widely used to compute the wealth index variable [28]. In DHS, wealth index is computed using a statistical procedure known as Principal Component Analysis (PCA) [28]. Typically, household wealth index is classified into population quintiles; poorest, poor, middle, rich and richest. In large household surveys like DHS where data on income and expenditure cannot be collected, wealth index is a reliable measure of the standard of living [29], has long been cited as a measure of Socioeconomic Position (SEP) of a household [30] and is widely used in the measurement of SEP related social inequalities. Educational status of the mother was categorized as no education, primary and secondary or higher education. Residence was classified as urban vs. rural, whereas gender was classified as male vs. female. Subnational region was classified into 18 subnational regions (S1 Table). The educational status and wealth have a natural ordering and are called ordered equity stratifiers whereas place of residence and regions are non-ordered equity stratifiers. Whether an equity stratifier is ordered or not affects the choice of summary measures to be calculated [22].

## Statistical analysis

As described above, using the 2019 version of the WHO HEAT application [23], the socioeconomic, gender and area-based inequalities in the IMR were analyzed. The WHO released the software in 2016 using free and publicly available R programming language and the R packages [24]. The motivation for the creation of the software application was to help researchers and decision makers gauge health disparities with standard approaches.

IMR was disaggregated by the dimensions of inequality as discussed above. In addition, summary measures of different use and statistical properties were adopted. Combination of absolute and relative, as well as simple and complex inequality summary measures was employed. These were Population Attributable Risk (PAR), Slope Index of Inequality (SII) and Ratio (R). The first two are absolute inequality measures and the last is a relative measure of inequality. While all measures were calculated for wealth and education equity stratifiers, only R and PAR were estimated for the gender, region and place of residence. That means, the computation of SII was restricted to education and wealth dimensions of inequality since it required an ordered equity stratifier [22, 23]

The type of health indicator of interest (favorable vs. adverse) and the inherent properties of the dimensions of inequality determine calculation and interpretations of summary measures for this inequality study [22, 23]. IMR is unfavourable indicator and calculation of summary measures for IMR is different from how we calculate them for favorable indicators.

R is a simple measure suitable for showing the relative differences between two categories within a dimension of inequality (i.e. urban vs. rural for residence). The other two (PAR, SII) are weighted complex measures of inequality that take into account sizes of subpopulations, thereby producing estimates reflective of the whole subpopulation size [22, 23]. R was calculated as estimates of IMR for one subgroup divided by that of another subgroup, where a subgroup in the numerator has higher IMR than a subgroup in the denominator. For instance, IMR in the poorest divided by in the richest was done to compute R for wealth dimension. While the PAR and SII are absolute measures, R measure relative inequality. Both PAR and SII take zero in the absence of inequality, and greater absolute values indicate higher levels of inequality. The SII becomes negative when IMR is disproportionately more concentrated among the poor and non-educated subgroups and positive when it is highly prevalent among the educated and relatively rich subgroups. The PAR is negative for unfavourable health indicators like IMR and indicates a higher burden of the IMR among the poorest and poor subgroups of wealth, non-educated and primary education groups, male sex, rural settings and Benguela subnational regions. R assumes a value of one when there is no inequality, with higher values indicating higher inequality. The detailed methods of calculation, interpretation and all other detailed properties of the measures employed in the study have been described elsewhere in detail [23].

To determine that the disparity in IMR is statistically significant between subgroups, a 95% CI was computed around point estimates. For absolute inequality measures, the lower and upper bounds of the CI must not include zero to declare that inequality exists. For the relative inequality measures, however, the interval must not contain one. Survey design specification was taken into account during analysis since the data came from complex sampling structure.

## Ethical issues

Since the analysis for the study relied on the dataset stored in the HEAT software application, there were no ethical barriers associated with the use of the data. Ensuring ethical procedures of the survey were the responsibilities of the institutions that implemented and led the survey

to ensure that the protocols are conform to international regulations for the protection of study participants.

## Results

Table 1 presents the distribution of IMR by the five dimensions of inequality and the population share of each category of the five dimensions of inequality. A total of 24,124 live births, born five years prior to the survey, were included. More than 61% of the births were in urban areas. Close to 31% were born to women with no education and 22% were born to women from the poorest subgroups. The largest proportion of the sample was drawn from Luanda region (29.3%).

The average IMR for Angola was close to 50 deaths per 1000 live births, lowest among the richest subgroup and highest among the poorest and poor (Fig 1 and Table 1). IMR was significantly lower among female infants compared to male infants. Large subnational regional variations existed; Benguela had the highest IMR whereas Moxico had the lowest (Table 1 and Fig 2). In terms of place of residence, children born in urban areas had lower chance of death than their rural counterparts (see Table 1).

**Table 1. IMR disaggregated by the five equity stratifiers, 2015 Angola demographic and health survey, Angola.**

| Dimensions of inequality | Categories | Point estimate of IMR (95% confidence interval) | Population |
|---|---|---|---|
| Wealth index | Poorest | 62.17 (53.3,72.5) | 5241 |
| | Poor | 63.72 (56.1, 72.3) | 5565 |
| | Middle | 46.99 (37.5, 58.8) | 5236 |
| | Rich | 42.36 (32.7, 54.8) | 4619 |
| | Richest | 22.76 (16.0, 32.3) | 3462 |
| Education | No-education | 49.59 (42.5, 57.8) | 7422 |
| | Primary | 61.56 (54.3, 69.7) | 9859 |
| | Secondary | 32.73 (26.5, 40.4) | 6843 |
| Place of residence | Rural | 61.17 (53.8, 69.5) | 9388 |
| | Urban | 42.52 (37.1, 48.7) | 14736 |
| Sex | Female | 42.95 (37.7, 49.0) | 12019 |
| | Male | 56.53 (50.5, 63.3) | 12105 |
| Subnational region | Cabinda | 26.63 (16.4, 43.0) | 458 |
| | Zaire | 35.37 (21.6, 57.3) | 498 |
| | Uige | 41.14 (27.9, 60.4) | 1385 |
| | Luanda | 31.51 (24.1, 41.1) | 7075 |
| | Cuanza Norte | 59.11 (45.3, 76.7) | 328 |
| | Cuanza Sul | 79.14 (61.9, 101.0) | 2022 |
| | Malanje | 38.46 (28.0, 52.7) | 956 |
| | Lunda Norte | 39.71 (25.1, 62.4) | 701 |
| | Benguela | 88.35 (73.7, 106.0) | 2326 |
| | Huambo | 61.06 (47.7, 77.9) | 1966 |
| | Bie | 52.69 (40.6, 68.1) | 1316 |
| | Moxico | 6.72 (2.5, 18.0) | 461 |
| | Cuando Cubango | 49.16 (34.7, 69.3) | 410 |
| | Namibe | 51.83 (42.6, 62.9) | 306 |
| | Huila | 66.73 (50.7, 87.3) | 2300 |
| | Cunene | 41.98 (28.1, 62.3) | 885 |
| | Lunda Sul | 32.28 (19.8, 52.3) | 473 |
| | Bengo | 23.31(12.9, 41.7) | 251 |

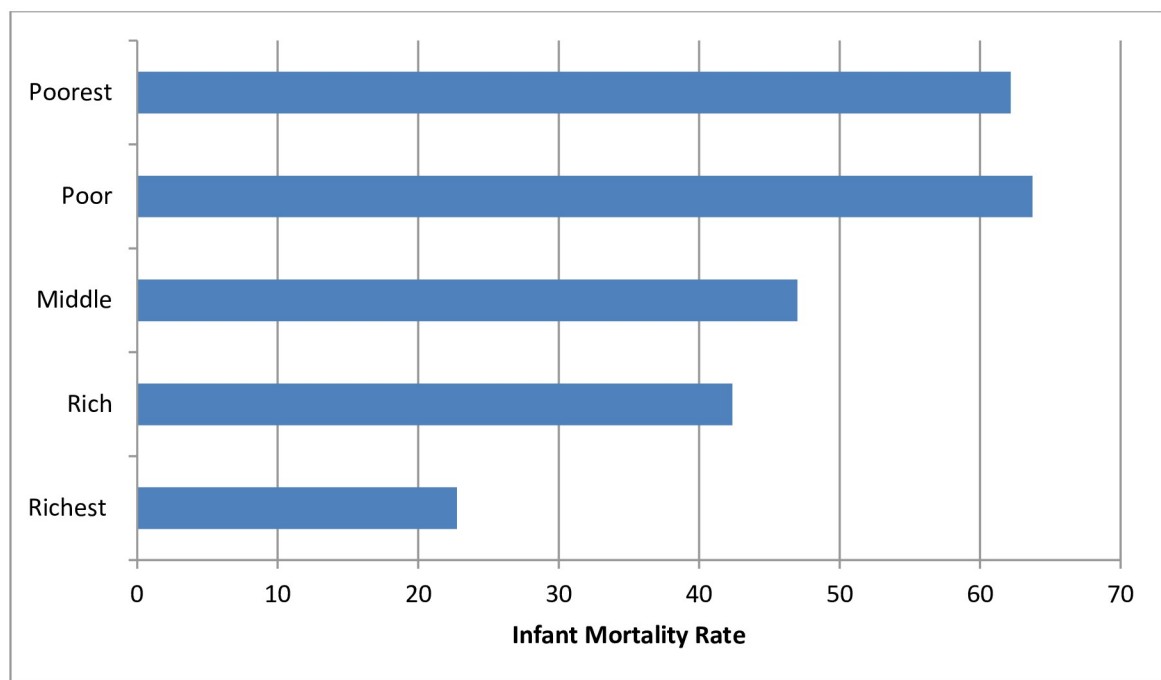

**Fig 1. IMR disaggregated by the wealth quintiles, Angola, 2015 ADHS.**

All the three measures of inequality in Table 2 indicate that wealth driven inequality in IMR remained in Angola to the detriment of children who fell towards the poorer end of the wealth quintile. Infants born to mothers in the poorest household wealth quintile experienced approximately 2 to 4 times more mortality. PAR showed that, the 2015 national figure for IMR would

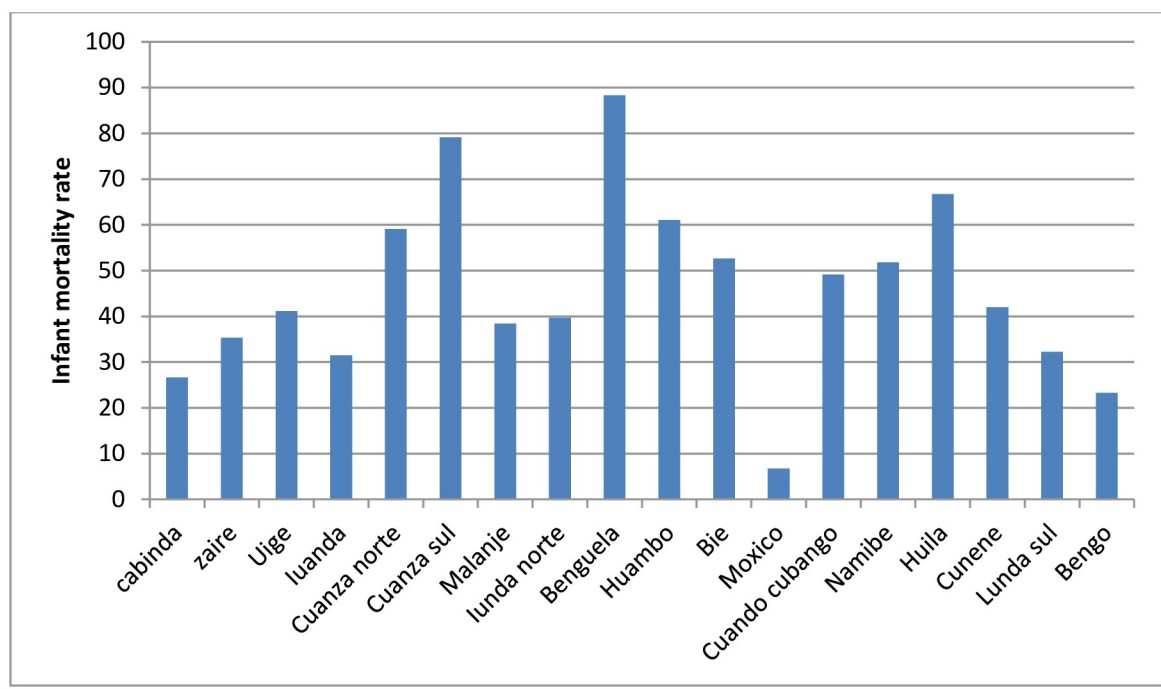

**Fig 2. IMR for the 18 subnational regions, Angola, 2015 ADHS.**

**Table 2. IMR inequality as shown by the different inequality measures across the five dimensions of inequality, 2015 ADHS.**

| Dimensions of inequality | Measures of inequality | Estimate (95%CI) |
|---|---|---|
| Wealth | R | 2.7(1.7, 3.8) |
| | SII | -47.0 (-56.8, -36.0) |
| | PAR | -27.0 (-28.4, -26.0) |
| Education | R | 1.5 (1.1,1.9) |
| | SII | -23.0 (-32.9, -12.0) |
| | PAR | -17.0 (-17.9, -16.0) |
| Place of residence | R | 1.4 (1.2,1.7) |
| | PAR | -7.3(-7.8, -6.7) |
| Gender | R | 1.3 (1.1, 1.5) |
| | PAR | -6.8 (-7.5, -6.2) |
| Subnational region | R | 13.1(-0.1, 26.4) |
| | PAR | -43.0 (-45.3, -4) |

R = Ratio; SII = Slope Index of Inequality; PAR = Population Attributable Risk.

have fallen by nearly 26 to 28 deaths per 1000 live births had the IMR among the poorest been reduced to the level in the richest quintile.

Similarly, the study revealed the glaring educational inequality in IMR. According to the R measure, infants born to illiterate women experienced 1.5 times (1.5; 95% CI: 1.1, 1.9) higher IMR than infants born to mothers who completed secondary or higher education.

The geographic disparity in IMR was substantial. The urban-rural differential of IMR was supported by both simple (R) and complex (PAR) summary measures. The PAR finding showed that the observed gap in the level of IMR between urban and rural settings was not as pronounced as the gap between the sub-national regions. In the absence of residence related disparity in IMR, the IMR in Angola in 2015 would have been decreased by 7 to 8 infant deaths per 1000 live births. The rate of reduction of the overall IMR in the country was even larger had the level of IMR in Benguela been reduced to the level in Moxico. See Table 2 for detail.

## Discussion

In this study, the state of inequality in IMR in Angola was examined across the different sub-population groups through the WHO HEAT software application. The magnitude of IMR differed substantially between population subgroups. The socioeconomic gradient in IMR occurred to the favour of infants born to wealthier and educated families. The geographic variations of IMR in the country were substantial, and male infants had a higher chance of death during the first year compared to female infants.

Development of strong equity-oriented interventions must be preceded and supported by high-quality evidence. This in-depth analysis of inequality of IMR in Angola following the best available methodology informs the creation of equity-oriented, meaningful interventions that decrease IMR disparity. The study highlighted that IMR inequality was present across the socioeconomic positions, between the urban and rural settings, as well as the subnational regions. This could lead to a conclusion that, the subgroups with the higher burden of infant mortality may receive little or no attention in the form of increasing coverage of interventions that would reduce the high infant mortality, and that they are lagging behind in terms of economy, education, and living standards. The SDG aims to reduce child and neonatal mortality rates to 25 and 12 deaths per 1000 live births, respectively, between 2015 and 2030 [21]. Given

substantial disparity of IMR between different subgroups as it currently stands, it will be challenging for Angola to attain the global target on child mortality unless attention is directed toward poorly performing subgroups.

The study highlighted that the national IMR figure masks the complete story of the problem in the country; some places have an unacceptably large clustering of IMR, sometimes reaching an average of 88 deaths per 1000 live births. Such geographical places continue to negatively influence the attainability of the 2030 SDG by directly affecting the national average IMR. Benguela region had the highest infant deaths whereas Moxico had IMR of fewer than 7 deaths per 1000 live births. Based on point estimate of the PAR measure, 43 fewer deaths per 1000 live births would have been observed in the country if the burden of infant death in the remaining regions were reduced to a level in Moxico region. This means that the present IMR in Angola could have been just 7 deaths, not 50 deaths per 1000 live births had subnational regional disparity not occurred.

The poor-rich disparity of IMR in Angola needs attention; the poor endure a disproportionately higher burden of infant mortality. If the IMR among the poorest was reduced to the level among the richest, the current national IMR would have been fallen by 27 deaths per 1000 live births, which translated to an IMR of just 23 deaths. Since under five mortality rate comprises of neonatal mortality rate (NMR) and IMR, the huge drop of IMR (from 50 to 23) could drive a significant move towards attainment of the global targets for child mortality even when NMR could not improve. Eliminating the poor-rich disparity has huge gains not just in terms of international agendas like SDG, but from a human right perspective, because all infants have the right to live irrespective of the socioeconomic positions of women and families they are born to. The present study compares well with available growing knowledge that wealth disparity could lead to disparity in IMR [13], and overcoming a wealth-driven gradient could translates to improved survival of infants. In low-and-middle income countries (LMICs), poorer households have a larger concentration of infant mortality [14]. Promoting pro-poor policies in LMICs would result in the reduction of inequalities in under five mortality, and governments in these countries need to consider distributing health facilities fairly with more attention drawn to the poor subgroups [31].

Similarly, the study confirmed sizeable educational inequality of IMR though not so severe as wealth-related inequality discussed above. Infants born to women who are not educated endure disproportionately greater risk of deaths. Also, it was possible in getting the national IMR to as low as 33 deaths (instead of the observed 50 deaths per 1000 live births) if all women in the sample were educated to the secondary level. In fact, maternal education has been shown to be positively associated with better health of infants [32]. For example, Kiross GT et al. (2019) showed that infants born to mothers who attended primary education had 28% lower odds of experiencing death compared with infants born to illiterate mothers, and a further 45% reduction if they were born to mothers who completed secondary education or higher [32].

Male infants had a higher chance of death than female infants and prior evidence supported the pro-female scenario of IMR [15, 16]. A possible explanation for a 'male disadvantage' in terms of death during the first year of life could be associated with male infant's higher risk of birth complication and infectious diseases [17]. Evidence also shows that male infants are more likely to be born premature, have relatively weaker immune system and tend to be more susceptible to infectious diseases such as syphilis, malaria, tetanus, and diarrheal diseases [33–35].

In the present study, nearly 7 fewer infant deaths per 1000 live births could have been recorded in Angola had infant deaths among males been reduced to a level among females. The findings call for the need to promote integration of gender sensitive and responsive

interventions into the country's reproductive health programs in order to reach both male and female babies equally and equitably. Further studies on the gender differential of infant mortality are important to guide interventions that would end male-female gap in the distribution of IMR. There may be other drivers that need to be targeted by appropriate interventions. However, exploring the drivers of gender differences in infant mortality is beyond the scope of the paper and this area warrants further investigation.

Residence driven disparity of IMR in Angola shows a greater number of infant deaths in rural settings compared with urban settings. The 50 deaths per 1000 live births in the country would have been lowered to 43 deaths if there were no urban-rural IMR disparity. The impact of residence related disparity of IMR on the realization of the 2030 SDG targets on child health should be the subject of further exploration. In a country such as Angola where birth rate is high, the deaths of hundreds of infants would be expected to which the presence of discernible inequality between urban and rural settings could contribute. Prior studies examining IMR variations according to place of residence also confirmed the higher concentration of infant deaths in rural and slum settings than in non-slum urban areas [36].

The study has strength. Since the study uses the high-quality WHO HEM database, the findings are reliable for decision making. However, the study suffers some limitations. The paper did not answer the question, "why did IMR inequality remain in Angola?" Future studies may carry out a decomposition analysis to disentangle the population variations in IMR of the different factors known to affect infant mortality.

## Conclusions

Survival advantage of infants differed greatly by the characteristics of the mother; infants born to socioeconomically strong woman and to woman who live in urban areas and some parts of the country (like Moxico) had better survivals rates. New interventions should focus on ensuring food security for the rural and poor residents in the country to alleviate poverty in order to eliminate the household wealth related differential of IMR. Similarly, increasing the proportion of females who complete secondary or more education could help eliminate the educational inequality of IMR. Finally, child survival interventions need to reach all children in the subnational regions with the highest clustering of infant mortality.

## Supporting information

**S1 Table. The 18 subnational regions (provinces) of Angola and their population size.** (DOCX)

## Acknowledgments

The author acknowledges the WHO for making the WHO HEAT software available for free for researchers based in low-income countries.

## Author Contributions

**Conceptualization:** Gebretsadik Shibre.

**Data curation:** Gebretsadik Shibre.

**Formal analysis:** Gebretsadik Shibre.

**Software:** Gebretsadik Shibre.

**Writing – original draft:** Gebretsadik Shibre.

**Writing – review & editing:** Gebretsadik Shibre.

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
