## [Decision Letter · Decision Letter 0]

11 Jun 2020

PONE-D-20-06660

Male infants and infants born to impoverished family suffer higher death toll in Angola: a nationally representative cross-sectional survey

PLOS ONE

Dear Dr. Shibre,

Thank you for submitting your manuscript to PLOS ONE. After careful consideration, we feel that it has merit but does not fully meet PLOS ONE’s publication criteria as it currently stands. Therefore, we invite you to submit a revised version of the manuscript that addresses the points raised during the review process.

We look forward to receiving your revised manuscript.

Kind regards,

Natasha McDonald

Associate Editor

PLOS ONE

Journal Requirements:

3. In your Methods section, please provide additional details regarding the dimensions of inequality, each group should be described sufficiently so that these analyses could be repeated.

4. We note you have included a table to which you do not refer in the text of your manuscript. Please ensure that you refer to Table 2 in your text; if accepted, production will need this reference to link the reader to the Table.

Reviewers' comments:

Reviewer's Responses to Questions

**Comments to the Author**

1. Is the manuscript technically sound, and do the data support the conclusions?

Reviewer #1: Yes

Reviewer #2: No

2. Has the statistical analysis been performed appropriately and rigorously? 

Reviewer #1: Yes

Reviewer #2: I Don't Know

3. Have the authors made all data underlying the findings in their manuscript fully available?

Reviewer #1: No

Reviewer #2: Yes

4. Is the manuscript presented in an intelligible fashion and written in standard English?

Reviewer #1: Yes

Reviewer #2: No

5. Review Comments to the Author

Reviewer #1: Dear Authors,

I have some comments for the article:

1. A literature review for similar studies on infant mortality conducted in other countries around the world is also needed. In the “Introduction” and “discussion” there should be more original studies instead of encyclopedias and institutional reports.

2. The purpose of the work is inappropriate and can be formulated, for example the aim of the study was to investigate the contribution of specific factors to social inequalities in infant mortality in Angola.

3. In the „Methods” in the date source section should mention the name of the country Angola

and the period of time over which inequalities were measured. Reference to the source of ADHS is also missing.

4. In the „Results” please add a table with descriptive statistics as well as a description of Table 2.

5. The use of word “survival” for the results of this article is inappropriate because the analysis concerning to mortality. Similarly, in the title of the paper there should be "mortality" instead of "death toll".

6. Please complete the keywords in the article, ie. education, poverty, rural – urban, and change “IMR” to “infant mortality” and “inequality” to “social inequalities in health”

7. Please put the changes with different color.

Reviewer #2: PLOS ONE manuscript review

Title- needs grammatical correction

English phrasing and grammar needs work throughout the manuscript. It is very difficult to read at present and there are multiple grammatical errors.

Introduction

Line 61 -no need for capitals.

- Sentence line 70: this is an obvious statement of fact. The line 'IMR could be an essential option' is confusing.

- expression of IMR would usually be per 1000 live births. Are there uncertainty ranges for international comparisons?

- Line 90: First and second reasons for justification seem the same? This paragraph would be better placed in the discussion.

- Line 96 does not make sense to me. Need more explanation of these terms rather than just a list.

- Line 99-Rationale for study. This is confusing. It would seem fairly self- evident that there will be within country variation of IMR. Perhaps a better way to phrase could be 'to gain an understanding of the factors driving in-country variability in IMR in Angola"

- Line 103-not sure why BY 1 is capitalized

- Need more background on Angola situation: why Angola? I What policy and other interventions already exist to address inequality in Angola, therefore what is known and remains unknown? Currently this study is not really justified by the introduction.

Methods

- Line 116-need reference for report

- The description of the ADHS is unclear-how many households were sampled? When was the data collected?

- Line 127 "inequality is measured for IMR" -this does not make sense. Please rephrase making clear the exposure and outcomes.

- Line 134-again this need to separate exposure & outcome.

- Line 142-listing all subnational regions is not required. Better to have a say supp file, perhaps with population size.

149 Did this use HEAT-Plus? More information needed about any transformations of the original data needed to use HEAT. Was all data in compatible format?

159 Line 162 "In addition, summary measures... - "this is unclear, possibly redundant given next line.

187-Ethical: as "data is stored in Heat software" it would be better to be clearer and more open about this earlier. My interpretation of this is that the author did not have to access the DHS data separately.

Results

Start with description of population-what was the sample size compared to overall population in Angola? What is the overall IMR? What is the range? How people/households were sampled in the DHS?

Table 1 is poorly laid out. Consider either better layout or maybe one or two key figures? For someone not familiar with the provinces of Angola, a heat map of the country may be a better way of presenting the geographic variation.

- Be consistent with reporting to one or 2 decimal places.

Overall, the results are extremely brief and don't help reader understand the data or its complexity.

Discussion

Should start with main finding.

- Not clear what the intersection between IMR and the HEAT indicators are: for example as IMR was highest in Benguela, did this correspond with poorer indicators in other areas?

6. PLOS authors have the option to publish the peer review history of their article (what does this mean?). If published, this will include your full peer review and any attached files.

Reviewer #1: No

Reviewer #2: Yes: Jane Elizabeth Hirst

---

## [Author Response · Author response to Decision Letter 0]

5 Aug 2020

Dear Dr Natasha McDonald, 

I have kindly appreciated the constructive feedback provided by you and the reviewers. Both you and the reviewers have signaled the importance of our study, together with valuable insights to better frame and clarify the message of the manuscript. I believe the revised manuscript has been significantly improved and the comments have been addressed adequately. 

Please find for your kind consideration the followings:

-A point-by-point response to the comments and suggestions (below).

-A new revised version of the manuscript with altered text highlighted

-A new revised clean version without track change. 

I hope that these changes meet with your favorable consideration. Please do not hesitate to get in touch if you require any further information.

Yours sincerely,

Gebretsadik Shibre

Editor’s comments 

 We suggest you thoroughly copyedit your manuscript for language usage, spelling, and grammar. If you do not know anyone who can help you do this, you may wish to consider employing a professional scientific editing service.

Response: Dear Dr, following your important suggestion, I now get the paper thoroughly copyedited by a native speaker. The name of the colleague who edited the paper is Dina Idriss-wheeler. 

3. In your Methods section, please provide additional details regarding the dimensions of inequality, each group should be described sufficiently so that these analyses could be repeated.

Response: I have now detailed that section following your important suggestion. 

4. We note you have included a table to which you do not refer in the text of your manuscript. Please ensure that you refer to Table 2 in your text; if accepted, production will need this reference to link the reader to the Table.

Response: Revised 

Reviewers' comments:

 Reviewer#1: Dear Authors,

I have some comments for the article:

1. A literature review for similar studies on infant mortality conducted in other countries around the world is also needed. In the “Introduction” and “discussion” there should be more original studies instead of encyclopedias and institutional reports.

Response: Dear reviewer, thank you for the important suggestion. I now thoroughly revised and updated the background, and situate it within the context of similar body of works on the same topic that have been carried out in different part of the world. I inserted journals to substantiate evidence extracted from other sources including reports. For some concepts, the evidence that I want to present in the paper has been found in the form of reports, and I therefore keep them in the manuscript but I also added other reference materials in both background and discussion sections. 

2. The purpose of the work is inappropriate and can be formulated, for example the aim of the study was to investigate the contribution of specific factors to social inequalities in infant mortality in Angola.

Response: Dear reviewer, the sole purpose of the paper was to provide an in-depth analysis of IMR disparity according to the various important dimensions of social inequality (wealth, education, residence, gender and regions). The paper aimed to first establish whether IMR disparity remains according to the five dimensions of inequality, since there is no study that comprehensively analyzed the extent of the IMR disparity in the country. One has to first confirm the existence of disparity before going further to explore underlying contributions of the same. Once we are aware that disparity exists around IMR based on the different equity startifers mentioned above, then we can do another study to examine the set of problems that would underlie the disparity using statistical procedure like decomposition analysis. Therefore, as far as my justification is concerned (the justification that you can find at the end of the introduction), this paper is of very relevant and timely for future high-level analysis. In any inequality study, the first step is knowing whether inequality exists in the first place, and descriptive studies like this current paper is enough to answer that research question. 

3. In the „Methods” in the date source section should mention the name of the country Angola and the period of time over which inequalities were measured. Reference to the source of ADHS is also missing.

Response: Revised 

4. In the „Results” please add a table with descriptive statistics as well as a description of Table 2.

Response: The description for Table 2 has now been added. Table 1 is presenting a descriptive statistics of the study. Table 1 is describing the level of IMR by different dimensions of inequality whereas Table 2 provides results of the summary measures for IMR INEQUALITY. 

5. The use of word “survival” for the results of this article is inappropriate because the analysis concerning to mortality. Similarly, in the title of the paper there should be "mortality" instead of "death toll".

Response: Revised. However, since mortality and survival are two sides of a coin, and to avoid overuse of the word mortality, I still used the word survival in some paragraphs. The use of “survival” in mortality studies is very common and has no problems.

6. Please complete the keywords in the article, ie. education, poverty, rural – urban, and change “IMR” to “infant mortality” and “inequality” to “social inequalities in health”

Response: Revised 

7. Please put the changes with different color.

Response: Revised 

Reviewer #2: PLOS ONE manuscript review

Title- needs grammatical correction

Response: Revised 

English phrasing and grammar needs work throughout the manuscript. It is very difficult to read at present and there are multiple grammatical errors.

Response: The language has been substantially improved. Pls see my response to the editor’s comment as well.

Introduction

Line 61 -no need for capitals.

- Sentence line 70: this is an obvious statement of fact. The line 'IMR could be an essential option' is confusing.

Response: Revised 

- expression of IMR would usually be per 1000 live births. Are there uncertainty ranges for international comparisons?

Response: Revised. Existing evidence provides only point estimates of IMR and I used them. 

- Line 90: First and second reasons for justification seem the same? This paragraph would be better placed in the discussion.

Response: I revised it. I now deleted some statements to avoid redundancy. 

- Line 96 does not make sense to me. Need more explanation of these terms rather than just a list.

Response: Revised 

- Line 99-Rationale for study. This is confusing. It would seem fairly self- evident that there will be within country variation of IMR. Perhaps a better way to phrase could be 'to gain an understanding of the factors driving in-country variability in IMR in Angola"

Response: Dear reviewer, based on a thorough literature review I carried out on the between population variability of IMR in Angola, I did not get good evidence on it. On the top of that, the WHO highly recommends inequality researchers to follow the WHO guideline to do inequality analysis. The recommendation is that, every inequality study has to be done with a mixture of both simple and complex, as well as relative and absolute summary measures. Plus, results from the chosen summary measures (Table 2 in this study) have to be presented after the disaggregated results (table 1 in this study). In the literature, however, there is no studies that meet this important criteria, but this paper has strictly followed that recommendation and thus has significantly contributed to what is already known on this matter. So, with apology, I have not accepted the suggestion “It would seem fairly self- evident that there will be within country variation of IMR. Perhaps a better way to phrase could be 'to gain an understanding of the factors driving in-country variability in IMR in Angola" since there is no rigorously conducted studies and hence no nuanced evidence on IMR variation by population groups. Dear Dr, the revised introduction (the last paragraph) delivers sufficient information about the novelty and the contribution of the paper (one contribution is in terms of methodology). The point, “'to gain an understanding of the factors driving in-country variability in IMR in Angola" is not the objective of this paper; this suggestion requires decomposition analysis. The current paper however aims to first examine whether IMR disparity exits across the 5 dimensions of inequality and if any, how much, using standard and internationally approved equity analysis techniques. For suggestion #2 for the first author, I also provided some similar response and you may find it useful for this current suggestion. The WHO handbook and a paper by WHO conducted using similar approaches are below: 

https://apps.who.int/iris/bitstream/handle/10665/164590/9789241564908_eng.pdf;jsessionid=AD00AFD3AA6BB5A7AD64DDA265B36C99?sequence=1

World Health Organization. Handbook on health inequality monitoring with a special focus on low and middle income countries [Internet]. Geneva: World Health Organization; 2013[cited 2019 Nov 18]. Available from: http://www.who.int/gho/health_equity/handbook/en/).

- Line 103-not sure why BY 1 is capitalized

- Need more background on Angola situation: why Angola? I What policy and other interventions already exist to address inequality in Angola, therefore what is known and remains unknown? Currently this study is not really justified by the introduction.

Response: I thank you for the very constructive feedback. The motivation of the study has now been substantiated with addition of new resources. The most important justification of this study is that, though evidence on IMR disparity is important for decision for Angola, this important knowledge is currently lacking. I kindly invite you to spend some time reading the last two paragraphs of the revised background. In one way or another, I incorporated your very useful suggestion.

Methods

- Line 116-need reference for report

- The description of the ADHS is unclear-how many households were sampled? When was the data collected?

Response: Revised adequately. 

- Line 127 "inequality is measured for IMR" -this does not make sense. Please rephrase making clear the exposure and outcomes.

Response: That section has undergone major revision following your comment. Since no regression model was run, there was no outcome and independent variables. I preferred to use the word “primary” to refers to IMR and “dimensions of inequality” to refers to the other five variables based on which IMR is to be disaggregated. 

- Line 134-again this need to separate exposure & outcome.

Response: Dear Dr, as I tried to explain above, there no is so-called outcome and exposure variable in the study. IMR is the one whose inequality is done, so this variable has come be known better as primary variable. The other variables, variables by which IMR was disaggregated, are known as dimensions of inequality. I revised the method that way. The sole reason being, I did not fit any predictive regression models, and the “outcome” and “exposure” terminologies should be confined to that kind of statistical situation. 

- Line 142-listing all subnational regions is not required. Better to have a say supp file, perhaps with population size.

Response: Thank you and I prepared a supplementary file that contains population size of each of the 18 provinces.

149 Did this use HEAT-Plus? More information needed about any transformations of the original data needed to use HEAT. Was all data in compatible format?

Response: Thank you for the important suggestion. The confusion around the use of HEAT has now been eliminated and additional details are made to the method to make it clear that the study uses HEAT application (the offline, standalone version), not HEA-plus.

159 Line 162 "In addition, summary measures... - "this is unclear, possibly redundant given next line.

Response: Revised 

187-Ethical: as "data is stored in Heat software" it would be better to be clearer and more open about this earlier. My interpretation of this is that the author did not have to access the DHS data separately.

Response: Thank you Dr and this confusion has now been revised. See above. 

Results

Start with description of population-what was the sample size compared to overall population in Angola? What is the overall IMR? What is the range? How people/households were sampled in the DHS?

Response: revised. The sampling procedure has been detailed under the “data source” section. The remaining comments are being incorporated in the result section. Since the HEAT application did not provide range (confidence interval) for the average national IMR, I just put the point estimate only. However, for the subgroup IMR in each dimensions of inequality, point estimate of IMR has been accompanied by the corresponding 95% confidence interval, and this is the main aim of the paper of the HEAT as well. 

Table 1 is poorly laid out. Consider either better layout or maybe one or two key figures? For someone not familiar with the provinces of Angola, a heat map of the country may be a better way of presenting the geographic variation.

Response: Dear reviewer, I cannot figure out the essence of “Table 1 is poorly laid out” since I chose the most simple table template that potentially every reader can understand. Its headings are labeled correctly. There is no barrier that prevents readers from understanding table 1. This is what researchers recommend. Further, since table is the best way to present both point estimates and the associated confidence interval together ( this what WHO recommends), I preserved table 1 as it is and two bar charts are now added to help better understand information contained in Table 1 following your important suggestion. Heat map, as you have rightly said, is important for presenting geographic variation. However, Heat Map cannot allow me to present 95% CI alongside point estimates, and without CI, interpreting point estimates alone is misleading. 

- Be consistent with reporting to one or 2 decimal places.

Overall, the results are extremely brief and don't help reader understand the data or its complexity.

Response: Following your important suggestion, I have now expanded the result, both through texts and graphs. Inconsistencies in reporting decimals have now been fixed. 

Discussion

Should start with main finding.

Response: Revised, I now brought the bottom line findings of the paper and used as an opening paragraph of the discussion section. 

- Not clear what the intersection between IMR and the HEAT indicators are: for example as IMR was highest in Benguela, did this correspond with poorer indicators in other areas?

Response: Dear reviewer, there is no so-called “HEAT indicators” in the study. The five variables, namely wealth, education, sex, residence and regions are called dimensions of inequality, or equity stratifiers. What I did in the study is that the magnitude of IMR was disaggregated by these dimensions, followed by summarizing the disparity through inequality measures. Since I did solely descriptive analysis, I cannot tell exactly why IMR was highest in one region but not in others, highest among the poorest but not among the richest subgroups, etc. The relationship of IMR with each of the five dimensions of inequality is assessed using the equity analysis techniques recommended by the WHO, and this examination is entirely descriptive. Further study may employ regression based analysis to establish their statistical relationship. However, for such high level analysis, this current study can serves as a foundation. Finally, though this is the limitation of the paper, I still tried to relate the observed disparity of IMR by dimensions of inequality with available knowledge and put some potential explanation why IMR differed by for instance education. See the revised discussion

---

## [Editor Report · Decision Letter 1]

25 Aug 2020

PONE-D-20-06660R1

Social inequality in infant mortality in Angola: evidence from a population based study

PLOS ONE

Dear Dr. Shibre,

Thank you for submitting your manuscript to PLOS ONE. After careful consideration, we feel that it has merit but does not fully meet PLOS ONE’s publication criteria as it currently stands. Therefore, we invite you to submit a revised version of the manuscript that addresses the points raised during the review process.

We look forward to receiving your revised manuscript.

Kind regards,

Jane Hirst

Academic Editor

PLOS ONE

Additional Editor Comments (if provided):

Thank you for your revision. The manuscript is much improved following your changes, however the following issues need to be addressed before it can be considered again for publication. The written English is better, although the paper is now very long in sections and could be more direct.

1. Abstract

Line 23: remove "evidence suggests that" at the start of the first sentence (these are unnecessary additional words).

Line 42: results: include actual numbers describing IMR and the ranges observed for the key outcomes

Line 48: conclusion: consider changing "remove" to "address"

2. Introduction

Line 64: Please do not use Encyclopaedia Britannica as a reference. IMR is clearly defined by WHO and others.

Overall the introduction needs to be shortened, with a focus just on the issues to be explored in this paper.

3. Methods

The statistical analysis is now much more thorough, but quite long for a journal article. Consider putting some of the more detailed information on calculation in a supplementary file.

Line 247: If I understand correctly, this could be more simply stated that SII was calculated for non-binary categorical variables (wealth and education only).

4. Results

Line 317: It would be helpful to give the range of IMR after the national figure in the text.

Table 1: Include "Point estimate of IMR (95% confidence interval) in the column heading. The subnational level statistics are the same as in the figure, so may be better just displayed in one place. If the figure is selected, the numbers and CI could move to supplementary material.

Table 2: Give a figure legend to describe R, SII and PAR as the table should be able to stand alone. Also, please explain in the results section how to interpret the negative SII and PAR values reported in table 2. For example, line 336 would be much easier to relate to the table if you used explained what these findings meant. For example you could state: Urban women had lower IMR than their rural counterparts, with an absolute risk difference of 1.4 deaths per 1000 live births (95%Ci 1.2-1.7). This difference increased once other factors were accounted for, with urban women suffering 7.3 per 1000 fewer deaths. (sorry I am not sure I have interpreted this correctly- this is the problem with the current presentation and lack of connection between the table and what is in the text. Clearly it is bad writing practice simply to state the table in works in the text, but for measures that are not obvious on how they should be interpreted, the author needs to give the reader more help.

Discussion: Line 472. Comparing the worse survival of male infants in Angola to female infants in patriacal countries such as India is false and misleading and I would removethrs section. it is implying that Angolan society favours females, and the information on male preference in other places is irrelevant here.

The discussion is very long. The paragraph starting at line 498 seems redundant and whilst raises points that are true, most of these points are covered elsewhere in the manuscript and this paragraph doesn't add to the message the author is trying to convey.

---

## [Author Response · Author response to Decision Letter 1]

25 Sep 2020

Dear Dr Jane Hirst, 

I am pleased that the paper has a publication potential once I have carried out the revisions requested by the journal. I have found the suggestions very useful and now, after a thorough revision, I believe that the paper has been sufficiently improved. I kindly appreciate your contribution to the furtherance of the paper via the constructive comments. Please do not hesitate to contact me should you think the manuscript would benefit from a further round of revision. 

Please kindly find the followings:

-A point-by-point responses to the comments raised (below)

-A clean version of the paper after accepting the responses to the comments

-A marked up copy of the paper with the responses reflected in track change

Regards, 

Gebretsadik Shibre

1. Abstract

Comment:

Line 23: remove "evidence suggests that" at the start of the first sentence (these are unnecessary additional words).

Response: Revised 

Comment:

Line 42: results: include actual numbers describing IMR and the ranges observed for the key outcomes

Response: Revised 

Comment:

Line 48: conclusion: consider changing "remove" to "address"

Response: Revised 

2. Introduction

Comment:

Line 64: Please do not use Encyclopaedia Britannica as a reference. IMR is clearly defined by WHO and others.

Response: Revised 

Comment:

Overall the introduction needs to be shortened, with a focus just on the issues to be explored in this paper.

Response: Dear Dr, I strongly believe that the entire introduction reflects different aspect of the issue explored in the paper. This current version of the introduction has been developed in response to the useful comments of the reviewers. I expanded based on their suggestions to do so in order to ensure every elements of a standard introduction needs to contain. Finally, still, I tried removing some statements. 

3. Methods

Comment:

The statistical analysis is now much more thorough, but quite long for a journal article. Consider putting some of the more detailed information on calculation in a supplementary file.

Response: Thank you Dr for highlighting another important area of improvement for the paper. I have now deleted the detail on methods of calculations of the measures after ensuring that it has been provided through a proper citation. I believe that even supplementary file containing this information is not necessary as the original document detailing this has been cited. 

Comment:

Line 247: If I understand correctly, this could be more simply stated that SII was calculated for non-binary categorical variables (wealth and education only).

Response: Dear Dr, that is not how we interpret SII. We cannot calculate SII for all non-binary variables. For instance, SII cannot be estimated for region. We always calculate this measure of inequality for dimensions of inequality that has natural ordering such as wealth and education. So, the statement I made in the method section regarding the SII calculation is correct. 

4. Results

Comment:

Line 317: It would be helpful to give the range of IMR after the national figure in the text.

Table 1: Include "Point estimate of IMR (95% confidence interval) in the column heading. The subnational level statistics are the same as in the figure, so may be better just displayed in one place. If the figure is selected, the numbers and CI could move to supplementary material.

Response: Dear Dr., the range of IMR for the national figure was not available in the HEAT software. The ranges were available for the subgroups only. That can be the limitation of the HEAT software, but in this paper, I cannot mention this issue as limitation of the paper since the aim of the paper was to show NMR disparity by the different subgroups for which range is available, not to show the disparity in NMR for the national figure. I just put the NMR for the national figure for discussion and comparison purpose, and also to accommodate suggestions of one or both of the reviewers of this paper. 

The statistics presented in the table is different from the one shown in the graphs. The standard method (as suggested by the HEAT) is to present the subnational statistics in a Table alongside the corresponding subpopulations, as I did exactly. Also, tables are the ideal place (not graphs/charts) to show 95% confidence. The charts used only for the selected dimensions of inequality( region and wealth) and are able to create a quick impression on the relative magnitude of NMR across the different subgroups of wealth and region. I used both table and charts to vary my data presentation techniques to improve the usability of the findings.

Comment:

Table 2: Give a figure legend to describe R, SII and PAR as the table should be able to stand alone. Also, please explain in the results section how to interpret the negative SII and PAR values reported in table 2. For example, line 336 would be much easier to relate to the table if you used explained what these findings meant. For example you could state: Urban women had lower IMR than their rural counterparts, with an absolute risk difference of 1.4 deaths per 1000 live births (95%Ci 1.2-1.7). This difference increased once other factors were accounted for, with urban women suffering 7.3 per 1000 fewer deaths. (sorry I am not sure I have interpreted this correctly- this is the problem with the current presentation and lack of connection between the table and what is in the text. Clearly it is bad writing practice simply to state the table in works in the text, but for measures that are not obvious on how they should be interpreted; the author needs to give the reader more help.

Response: Thank you for point this important issue out.

I have now explained what these abbreviations mean underneath the table. I clearly explained in the method section how the summary measures should be interpreted. The perfect place to explain the calculations and interpretations of the summary measures is the methods section and I have now done that. Both the table and texts must be interpreted in accordance with what has been presented in the method section. See my revisions on the method section.

Comment:

Discussion: Line 472. Comparing the worse survival of male infants in Angola to female infants in patriacal countries such as India is false and misleading and I would remove thrs section. it is implying that Angolan society favours females, and the information on male preference in other places is irrelevant here.

The discussion is very long. The paragraph starting at line 498 seems redundant and whilst raises points that are true, most of these points are covered elsewhere in the manuscript and this paragraph doesn't add to the message the author is trying to convey.

Response: Thank you, I revised it.

---

## [Editor Report · Decision Letter 2]

8 Oct 2020

Social inequality in infant mortality in Angola: evidence from a population based study

PONE-D-20-06660R2

Dear Dr. Shibre,

We’re pleased to inform you that your manuscript has been judged scientifically suitable for publication and will be formally accepted for publication once it meets all outstanding technical requirements.

Kind regards,

Jane Hirst

Guest Editor

PLOS ONE

Additional Editor Comments (optional):

Thank you for your revised submission. I'm pleased to let you know that the article is suitable for publication.
---

## [Editor Report · Acceptance letter]

12 Oct 2020

PONE-D-20-06660R2 

Social inequality in infant mortality in Angola: evidence from a population based study 

Dear Dr. Shibre:

I'm pleased to inform you that your manuscript has been deemed suitable for publication in PLOS ONE. Congratulations! Your manuscript is now with our production department. 

Kind regards, 

on behalf of

Dr. Jane Hirst 

Guest Editor

PLOS ONE